# Single cell analysis reveals immune cell–adipocyte crosstalk regulating the transcription of thermogenic adipocytes

Prashant Rajbhandari[1,2†]*, Douglas Arneson[3,4†], Sydney K Hart[2], In Sook Ahn[3], Graciel Diamante[3,4], Luis C Santos[2], Nima Zaghari[4], An-Chieh Feng[5], Brandon J Thomas[5], Laurent Vergnes[6], Stephen D Lee[1], Abha K Rajbhandari[7], Karen Reue[6,8], Stephen T Smale[5,8], Xia Yang[3,4,8], Peter Tontonoz[1,8,9]*

[1]Department of Pathology and Laboratory Medicine, University of California, Los Angeles, Los Angeles, United States; [2]Diabetes, Obesity, and Metabolism Institute, Icahn School of Medicine at Mount Sinai, New York, United States; [3]Department of Integrative Biology and Physiology, University of California, Los Angeles, Los Angeles, United States; [4]Bioinformatics Interdepartmental Program, University of California, Los Angeles, Los Angeles, United States; [5]Department of Microbiology, Immunology, and Molecular Genetics, University of California, Los Angeles, Los Angeles, United States; [6]Department of Human Genetics, David Geffen School of Medicine, University of California, Los Angeles, Los Angeles, United States; [7]Department of Psychiatry and Neuroscience, Icahn School of Medicine at Mount Sinai, New York, United States; [8]Molecular Biology Institute, University of California, Los Angeles, Los Angeles, United States; [9]Department of Biological Chemistry, University of California, Los Angeles, Los Angeles, United States

*For correspondence:
prashant.rajbhandari@mssm.edu (PR);
ptontonoz@mednet.ucla.edu (PT)

†These authors contributed equally to this work

**Abstract** Immune cells are vital constituents of the adipose microenvironment that influence both local and systemic lipid metabolism. Mice lacking IL10 have enhanced thermogenesis, but the roles of specific cell types in the metabolic response to IL10 remain to be defined. We demonstrate here that selective loss of IL10 receptor α in adipocytes recapitulates the beneficial effects of global IL10 deletion, and that local crosstalk between IL10-producing immune cells and adipocytes is a determinant of thermogenesis and systemic energy balance. Single Nuclei Adipocyte RNA-sequencing (SNAP-seq) of subcutaneous adipose tissue defined a metabolically-active mature adipocyte subtype characterized by robust expression of genes involved in thermogenesis whose transcriptome was selectively responsive to IL10Rα deletion. Furthermore, single-cell transcriptomic analysis of adipose stromal populations identified lymphocytes as a key source of IL10 production in response to thermogenic stimuli. These findings implicate adaptive immune cell-adipocyte communication in the maintenance of adipose subtype identity and function.
DOI: https://doi.org/10.7554/eLife.49501.001

## Introduction

Adipose tissue plays an important role in the maintenance of energy balance in mammals. A chronic imbalance between energy intake and expenditure increases adiposity and leads to obesity and pre-disposes to the development of metabolic disease (*Rosen and Spiegelman, 2014*; *Wajchenberg, 2000*). White adipose tissue (WAT) is essential for triglyceride (TG) storage, whereas thermogenic brown adipose tissue (BAT) dissipates energy as heat through mitochondrial uncoupling mechanisms including uncoupling protein 1 (UCP1) (*Cannon and Nedergaard, 2004*;

*Chechi et al., 2013*; *Saely et al., 2012*). A subpopulation of mature adipocytes in certain WAT depots, known as beige adipocytes, also acquires thermogenic capacity in response to cold or hormonal stimuli. Increasing the activity of thermogenic adipocytes in animal models counteracts the development of obesity and diabetes (*Cohen et al., 2014*; *Harms and Seale, 2013*; *Song et al., 2016*; *Villanueva et al., 2013*). Therefore, a better understanding of the mechanisms controlling adipose thermogenesis could inform the development of new therapies for metabolic diseases.

In times of metabolic need, white adipocytes mobilize stored lipids through hydrolysis of triglycerides and release of free fatty acid (*Zechner et al., 2009*). Activation of β-adrenergic receptors (β-ARs) by catecholamines released by the sympathetic nervous system (SNS) is a major physiological inducer of lipolysis and adipose thermogenesis (*Bartness et al., 2014*; *Harms and Seale, 2013*; *Vitali et al., 2012*; *Wang et al., 2013*). Cold sensation triggers β-AR activation in adipose tissue, resulting in cAMP production and the activation of protein kinase A (PKA) and p38 kinase. These kinases initiate phosphorylation cascades that impinge on C/EBP, ATF, and CREB transcription factors that regulate the expression of gene involved in mitochondrial biogenesis (such as *Ppargc1*) and thermogenesis (such as *Ucp1*) (*Harms and Seale, 2013*). Adrenergic stimuli induce lipolysis in adipocytes through PKA-dependent phosphorylation of lipases including hormone-sensitive lipase (HSL) and adipose triglyceride lipase (ATGL) (*Duncan et al., 2007*; *Jaworski et al., 2007*; *Zechner et al., 2009*). Free fatty acid (FA) released from adipocytes can be used by peripheral tissues for mitochondrial respiration. The importance of β-AR-dependent lipid mobilization and thermogenesis is underscored by the observations that genetic deletion of β-ARs or adipocyte-specific loss of ATGL causes a drastic reduction in energy expenditure that predisposes mice to obesity (*Bachman et al., 2002*; *Schreiber et al., 2017*).

Clearance of catecholamine is critical in terminating β-adrenergic signals, and local catabolism of catecholamine is an established mechanism for negative regulation of adrenergic signaling in adipose tissue (*Camell et al., 2017*; *Pirzgalska et al., 2017*; *Song et al., 2019*). However, our prior studies revealed an unexpected role for IL10 signaling in the inhibition of β-adrenergic signaling and adipose thermogenesis (*Rajbhandari et al., 2018*). Mice globally-deficient in IL10 expression have enhanced energy expenditure and are protected from diet-induced obesity. We showed that chromatin accessibility at thermogenic genes in subcutaneous adipose tissue was linked to IL10 signaling; however, whether adipocytes themselves are the primary target for the metabolic effects of IL10 in vivo remains to be established. Here, we show that adipocyte-specific deletion of IL10Rα increases adipose adrenergic signaling, recapitulating the effects of whole-body IL10 deletion. Single nuclei transcriptomics of mature subcutaneous adipocytes identified a thermogenic adipocyte subtype that was enriched in IL10Rα-deficient mice. Additionally, single-cell RNA-sequencing (scRNA-Seq) of adipose stromal populations revealed an increase in IL10-producing adaptive immune cells under adrenergic stimulation. These results define an immune-adipocyte axis that plays an important role in the modulation of the adipose adrenergic response.

## Results

### Loss of AdIL10Rα in adipocytes promotes thermogenesis and confers obesity resistance

We previously reported that global IL10-deficient mice have increased energy expenditure and browning of white adipose tissue (*Rajbhandari et al., 2018*). To definitively determine whether mature adipocytes are the major cellular targets for these metabolic effects of IL10 we generated adipocyte-specific IL10 receptor α-deficient mice (AdIL10Rα KO) by crossing *Il10ra*^FL/FL mice to *Adiponectin-Cre* transgenics. Prior published studies have reported that *Adiponectin-Cre* transgenic mice do not present an obvious metabolic phenotype and therefore we chose to use *Cre*-negative floxed mice as the controls for our studies (*Kong et al., 2014*; *Villanueva et al., 2013*).

We confirmed loss of IL10Rα in adipocytes from AdIL10Rα KO mice by western blotting (*Figure 1A*). There was no difference in body weight or food intake between 10 week-old chow-fed AdIL10Rα KO mice and floxed littermate controls (*Figure 1—figure supplement 1A and B*). However, metabolic cage analysis revealed increased oxygen consumption and energy expenditure in AdIL10Rα KO mice, without changes in locomotion and body mass (*Figure 1B* and *Figure 1—figure supplement 1C*). To determine if this difference in energy expenditure was associated with

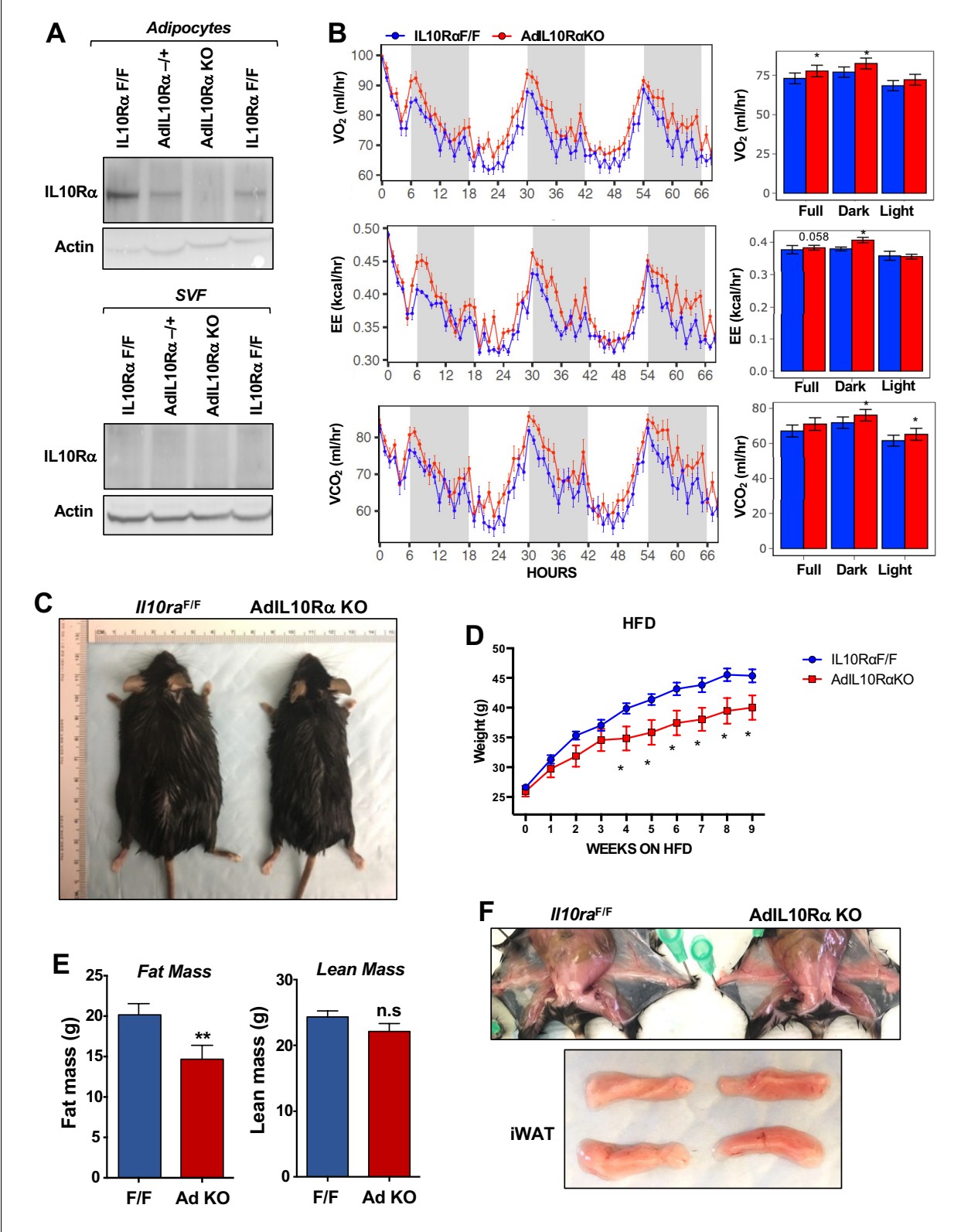

**Figure 1.** Ablation of IL10Rα in adipocytes increases energy expenditure and protects mice from diet-induced obesity. (A) Immunoblot analysis of IL10Rα expression in SVF and adipocyte fractions of iWAT from chow-fed 10 week-old mice. (B) Oxygen consumption (VO2, ml/hr), energy expenditure (EE, kCal/hr), and carbon dioxide production (VCO2, ml/hr) of chow-fed 10-week-old IL10RαF/F and AdIL10Rα KO mice were analyzed in Columbus Oxymax metabolic chambers (12 hr light/dark cycles, 72 hr total duration, each light/dark bar represents 12 hr duration). N = 9,7. Statistical analysis was

*Figure 1 continued on next page*

*Figure 1 continued*

performed using ANCOVA. (**C**) External appearance of representative 9 week HFD-fed mice. (**D**) Body weight of mice fed chow diet for 10 weeks and then 60% high-fat diet (HFD) for 9 weeks. N = 13, 7. (**E**) Fat and lean mass of mice in D fed HFD for 6 weeks. (**F**) Internal and gross appearance of iWAT from representative IL10RαF/F and AdIL10RαKO mice. *, p<0.05.

DOI: https://doi.org/10.7554/eLife.49501.002

The following figure supplement is available for figure 1:

**Figure supplement 1.** AdIL10Rα KO mice are protected against diet-induced obesity.

DOI: https://doi.org/10.7554/eLife.49501.003

protection against diet-induced obesity, we challenged AdIL10Rα KO with high-fat diet (60% calories from fat). After 8–9 weeks of HFD feeding, both male and female AdIL10Rα KO mice were protected against weight gain compared to floxed littermate controls (*Figure 1C,D* and *Figure 1—figure supplement 1D*). MRI analysis of body composition showed that the difference in body weight was entirely due to a change in fat mass (*Figure 1E*). Consistent with these findings, the inguinal WAT (iWAT) of AdIL10Rα KO mice was visibly redder than that of controls, suggesting increased tissue 'browning' (*Figure 1F*).

To test the influence of the IL10-IL10R axis on adipose adrenergic responses, we exposed AdIL10RαKO and floxed control mice to cold stress (4°C) for 24 hr. Analysis of iWAT gene expression by real-time PCR showed increased expression of *Ucp1*, *Elovl3*, *Ppargc1* and other thermogenic genes, but no change in general adipose markers such as *Fabp4* and *Pparg* in AdIL10Rα KO mice (*Figure 2A*). Similar results were observed in AdIL10Rα KO mice treated with β3-adrenergic agonist (CL 316,243; CL, 1 mg/kg/day for 4 days; *Figure 2B*). To gain insight into the global adipose gene expression changes in AdIL10Rα KO mice, we performed RNA-seq on iWAT. We identified 214 genes that were enriched more than 1.5-fold in AdIL10Rα KO mice compared to control mice (presented as a heatmap as a function of percentile expression in *Figure 2C*). The data revealed a selective increase in the thermogenic gene program in AdIL10Rα KO mice compared to controls. The gene expression differences between AdIL10Rα KO mice and controls were highly consistent with those observed in global IL10-deficient mice compared to WT controls (*Rajbhandari et al., 2018*), strongly suggesting that the effects of IL10 on adipose tissue gene expression are mediated predominantly through direct action of IL10 on adipose IL10Rα. These data also supporting a specific inhibitory effect of IL10Rα signaling on adrenergic-responsive pathways. We also noted that several genes that were more highly expressed in control mice compared to AdIL10Rα KOs have been linked to negative regulation of thermogenesis. For example, *Cnotl1* and *Brd2* have been reported to negatively regulate the mRNA stability and transcription of UCP1, respectively (*Figure 2C*) (*Takahashi et al., 2015*). In support of the calorimetric findings, we found increased mitochondrial respiration in the iWAT of AdIL10Rα KO mice compared to controls by Seahorse assays (*Figure 2D*).

## Identification of thermogenic adipocytes by SNAP-Seq

The data above show that IL10 acts directly on adipocyte AdIL10Rα to regulate the thermogenic gene program in adipocytes. To further dissect the role of the IL10-IL10R axis in regulating the identity and physiology of mature adipocytes, we performed single-cell analyses. As there were no prior reports of single primary adipocyte transcriptomics, we optimized a <u>S</u>ingle <u>N</u>uclei <u>A</u>dipocyte RNA sequencing approach (SNAP-seq) for assessing gene expression in mature adipocytes derived from mouse iWAT (*Figure 3A* and see Materials and methods). The critical step in this procedure is the isolation and purification of adipocyte nuclei which overcomes technical obstacles related to the handling of lipid-laden adipocytes. The single nuclei suspension (n ~ 10,000) was subjected to snRNA-Seq using the 10XGenomics platform, and libraries were sequenced with dedicated 400 million reads per sample (*Figure 3A*).

We chose to analyze mice exposed to a 24 hr cold challenge in these initial studies in order to increase our chance of identifying thermogenic adipocyte populations. We used the 10X genomics data processing and analysis platform to generate cell clusters and identities (see Materials and methods). To classify the adipocyte populations based on gene expression, we performed cluster analysis as represented by t-distributed stochastic neighbor embedding (t-SNE) plots. Remarkably, this analysis revealed that the adipocytes from iWAT of chow-fed C57BL/6 mice were highly

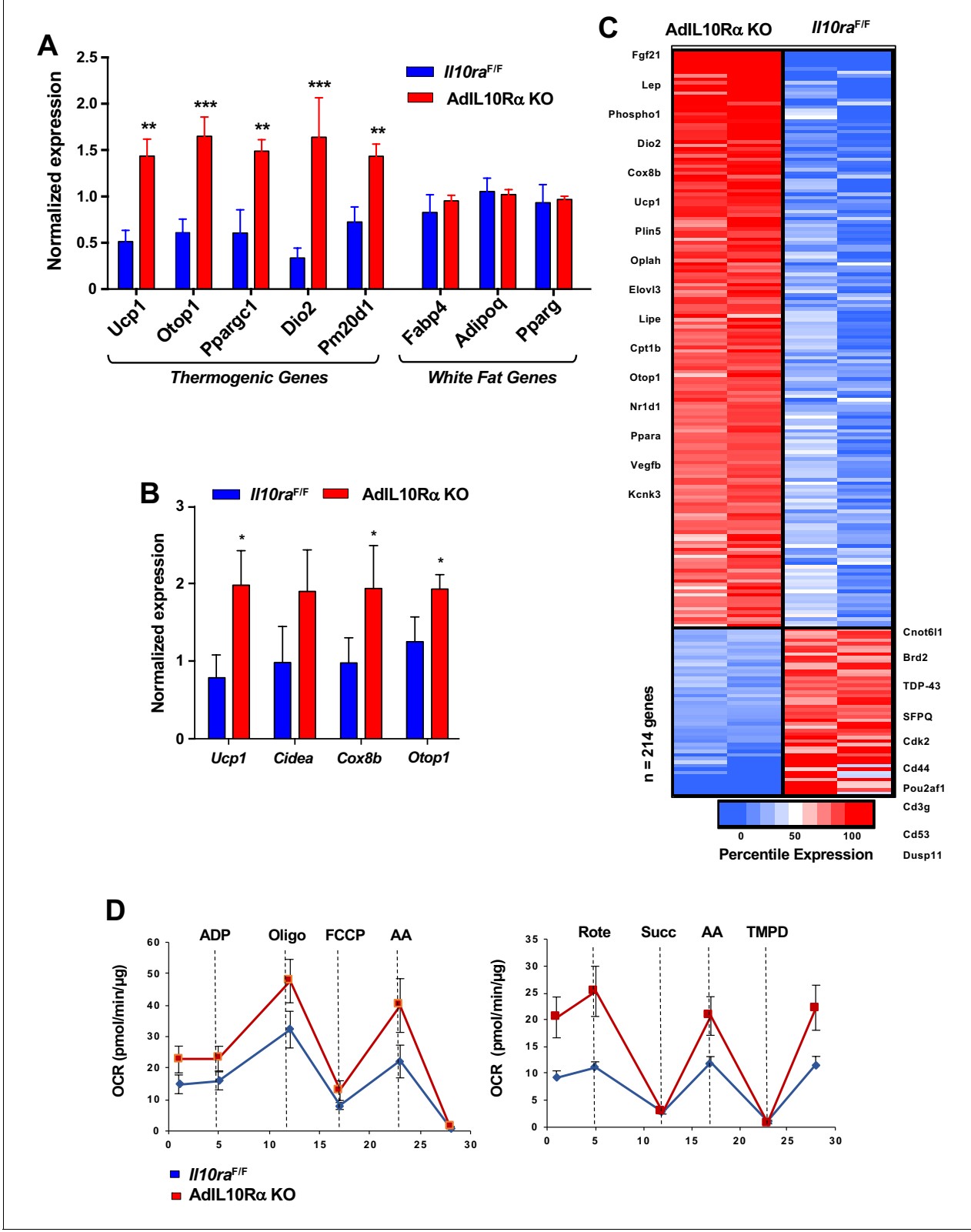

**Figure 2.** IL10R deficiency promotes adipose tissue browning. (**A and B**) Real-time PCR analysis of gene expression in iWAT from 10 week 24 hr cold-exposed (**A**) or CL 1 mg/kg/day for 4 days; **B**) IL10RαF/F and AdIL10RαKO mice. N = 5,5. *, p<0.05; **, p<0.01; ***, p<0.0001. (**C**) Heatmap representation of genes that changed >1.5 fold (p-value<0.01) as a function of percentile expression by RNA-Seq of iWAT from 10 week-old 24 hr cold-*Figure 2 continued on next page*

*Figure 2 continued*

exposed IL10Rα and AdIL10Rα KO mice. Genes are grouped as upregulated (Red) or downregulated (Blue). (**D**) Average oxygen consumption rate (OCR) in coupling (left) and electron flow (right) assays of mitochondria isolated from iWAT of mice in (**A**).

DOI: https://doi.org/10.7554/eLife.49501.004

heterogeneous. We were able to distinguish 14 distinct clusters (*Figure 3B*). The violin plots in *Figure 3C* further revealed that each cluster also uniquely express marker genes that were preferentially expressed in individual cluster. The tSNE-plot in *Figure 3D* further show localized expression of genes in particular cluster. Among all the clusters, we noticed a high enrichment of genes involved in fatty acid metabolism such as *Adrb3* and *Acsl1* in cluster 9. Furthermore, genes encoding β3-AR, HSL, and ATGL were highly overrepresented in this cluster, as were a variety of beige/brown adipocyte markers (*Figure 3E*). The gene expression profile of Type nine adipocytes was indicative of a highly metabolically active population whose characteristics were potentially consistent with thermogenic 'beige' adipocytes.

To address the possibility that contaminating stromalvascular cells might have given rise to one or more of these clusters we performed a Fisher's exact test between pairwise sets of cell type marker genes (determined by unadjusted and adjusted p-value<0.05) to find cell types that had substantial overlaps in their marker genes (denoting transcriptional similarity). Cell types from both mature adipocyte nuclei and stromal vascular fraction (SVF) single cells were used in this analysis and they were grouped using hierarchical clustering with tiles colored by -Log10 Bonferroni adjusted p-values. Adjusted p-values were thresholded to aid in visualization with values less than $10^{-5}$ set to $10^{-5}$. As shown in the diagonal correlation in *Figure 3—figure supplement 1*, we did not find high degree of transcriptional similarity between SVF and adipocyte clusters (top). However, under stringent p-value adjustment, the transcriptomic state of adipocyte clusters 12 and 14 correlated with markers of adaptative immune cells (bottom). Thus, we cannot exclude the possibility that clusters 12 and 14 may be contaminated with immune cells.

## Identification of a cold-responsive thermogenic 'beige' adipocyte population by SNAP-seq

To further test the hypothesis that Type nine adipocytes were the thermogenic beige population, we subjected mice to different thermogenic conditions, including cold stress (4°C) for 24 hr, 48 hr, and 4 days, or treated them with CL for 4 days (1 mg/kg/day). We then performed SNAP-seq on adipocytes derived from iWAT as described above. We performed unbiased aggregated clustering of the processed data for all the conditions as a tSNE-plot (*Figure 4A*). The aggregated cluster represents ~54,000 cell and allows us to confidently assign biological function to each cluster. Hence, to infer the biological properties of the cells in each cluster, we performed cell-type pathway enrichment analysis (Gene ontology (GO), KEGG, Reactome, and BIOCARTA) (*Supplementary file 1*) using enriched genes in each cluster based on false discovery rate (FDR) set at <0.05. This analysis revealed that different adipocyte subtypes express distinct genes with important roles in adipose tissue development, insulin signaling, hypoxia signal, inflammation, lipid synthesis and transport, angiogenesis, myogenesis, hormone responses, mitochondrial respiration, and fatty acid metabolic process (shown in *Supplementary file 1*). Type one adipocytes appeared to represent classical adipocytes and they expressed genes associated with adipose development, lipid responses, the insulin pathway, and response to corticosteroids (e.g. *Fto, Vldlr, Insr, Apod, Klf9, Sh3pxs2b*); Type three adipocytes were enriched for genes involved in blood vessel morphogenesis and angiogenesis; both Type 3 and 14 adipocytes were enriched for genes involved glycolysis. Type 6 and 7 adipocytes were enriched for genes involved in muscle metabolic process and myogenesis; Type 10 adipocytes were enriched for genes involved in the immune response; Type 11 adipocytes were enriched for cell cycle genes; Type 14 adipocytes were enriched for genes involved in mitochondrial ATP synthesis and respiration. Some adipocyte clusters displayed more commonality in gene expression with others and shared similar biological process, such as the abundant Type 1, 2, 3 and 7, 6 and 17 adipocytes (*Supplementary file 1*).

The top five enriched pathways determined by FDR from each cell type cluster were selected (from *Supplementary file 1*) and only unique pathways were kept (some top enriched pathways

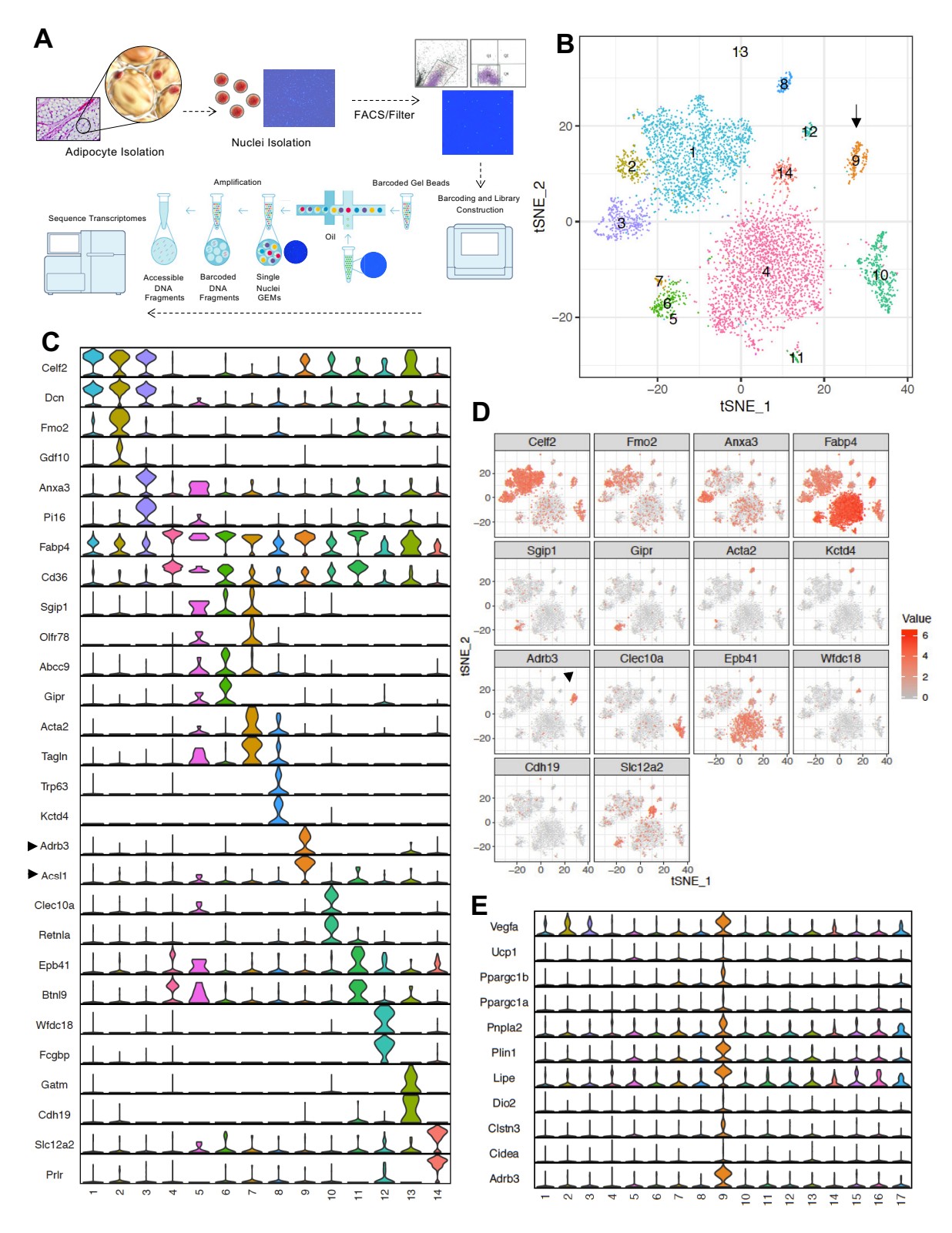

**Figure 3.** SNAP-seq reveals heterogeneity of tissue adipocytes from iWAT. (**A**) Workflow showing DAPI-stained nuclei pre- and post-FACS/filtration that underwent microfluidic partitioning and library preparation in the 10X genomics platform followed by sequencing using an Illumina HiSeq 4000. (**B**) tSNE-plot showing 14 clusters from ~6000 adipocytes derived from iWAT of mice exposed to cold for 24 hr. Each colored dot is an adipocyte assigned to a cluster based on transcriptomic signature. (**C**) Normalized expression values of the top two adipocyte subtype-specific cluster genes from (**B**)
*Figure 3 continued on next page*

*Figure 3 continued*

plotted as violin plots with clusters as rows and genes as columns. (D) tSNE-plot showing cluster-specific expression of selected marker genes from (C). (E) Normalized expression values of indicated genes in subtype-specific clusters plotted as violin plots with clusters as rows and genes as columns. Black arrow is pointing toward metabolically active Type nine adipocyte cluster and enriched gene.

DOI: https://doi.org/10.7554/eLife.49501.005

The following figure supplement is available for figure 3:

**Figure supplement 1.** Multimodal analysis of SVF and adipocyte single-cell sequencing data.

DOI: https://doi.org/10.7554/eLife.49501.006

were shared across cell type clusters). Enriched pathways (rows) were clustered with hierarchical clustering. The size of each dot represents the -log10 FDR of the pathway enrichment and the color of each dot corresponds to the fold enrichment of each pathway (red- higher enrichment, gray- lower enrichment). In *Figure 4—figure supplement 1*, we report the top five scoring pathways for each cell type cluster across all pathway sources (*Figure 4—figure supplement 1A*) or from KEGG pathways (*Figure 4—figure supplement 1B*). Overall, the most striking cluster was the Type nine adipocytes. Pathways enriched for triglyceride and neutral lipid catabolism, hormone-sensitive lipase (HSL)-mediated triglyceride hydrolysis and PPARα signaling were particularly enriched in type nine cluster.

Among all clusters (1-17), we noticed a very distinct sub-clustering of Type nine adipocytes, with a gradient of adipocyte subtypes in this population from room-temperature (RT) housed mice (top-RED), to mice exposed to cold for 4 days (Middle-Blue), to mice treated with CL (Bottom-Pink) (*Figure 4B*). Consistent with the data in *Figure 3*, the genes encoding β3-AR, HSL, ACSL1 were selectively enriched in the Type nine cluster (*Figure 4C* and *Figure 4—figure supplement 2A,B*). The tSNE-plot in *Figure 4D* reveals localized expression of thermogenic genes in cluster 9. Cluster nine adipocytes from CL- and cold-treated mice showed expression of markers of brown genes such as *Ucp1, Ppargc1a, Cidea, Dio2,* whereas *Adrb3, Lipe, Vegfa, Pnpla2* were expressed by most of the adipocytes in cluster 9 regardless of thermogenic stimuli (*Figure 4D*). Violin plots of thermogenic genes further showed upregulation of genes such as *Ucp1, Ppargc1a, Ppargc1b,* and *Cidea* in Type nine adipocytes in mice treated with thermogenic stimuli, confirming these cells as a bona-fide 'thermogenic adipocyte' cluster (*Figure 4E*). By contrast, genes involved in lipid mobilization such as *Adrb3 and Lipe* were abundant in all Type 9 cells regardless of stimulus, underscoring the relevance of these genes in thermogenic responsiveness and FFA metabolism. *Il10, Il10ra,* and *Il10rb* expression showed a relatively even distribution among mature adipocyte populations (*Figure 4—figure supplement 2C*).

We further confirmed the presence of Type nine adipocytes and validated co-expression of thermogenic genes by using RNAscope fluorescence in situ hybridization (FISH) in iWAT of mice treated with saline or CL (see Materials and method section). As shown in *Figure 4F*, CL treatment caused a robust increase in the expression of *Ucp1, Pppargc1b,* and *Adrb3* compared to control. Moreover, these transcripts showed a striking co-localization in only a set of adipocytes. We speculate that this population corresponds to Type nine adipocytes.

## Increased abundance of metabolically active type nine adipocytes in AdIL10Rα KO mice

To probe if IL10 signaling influenced the percentage of thermogenic adipocytes, we exposed both floxed-control and AdIL10Rα KO mice to either cold stress (4°C) for 24 hr, 48 hr, or 4 days, or treated them with CL (1 mg/kg/day) for 4 days, and performed SNAP-seq on iWAT of these mice. Cluster nine from the aggregated data showed a progressive increase in the percentage of cells upon exposure to cold for increasing lengths of time (3.4% to 5.7% to 12.1% from 24 hr to 48 hr to 4 days at 4°C, respectively) as shown both by tSNE- and dot-plots and treatment of mice with CL even more dramatically increased the abundance of Type 9 cells to 23.5% (*Figure 5A*-top and bottom). Dot-plotting further showed that among all clusters, type 1, 4, and nine showed the most changes under the different conditions, and among them only type nine adipocytes showed an adrenergic-dependent positive increase in cell fractions (*Figure 5A*-bottom). We next compared data from control and AdIL10Rα KO mice, and in agreement with the whole tissue RNA-seq data

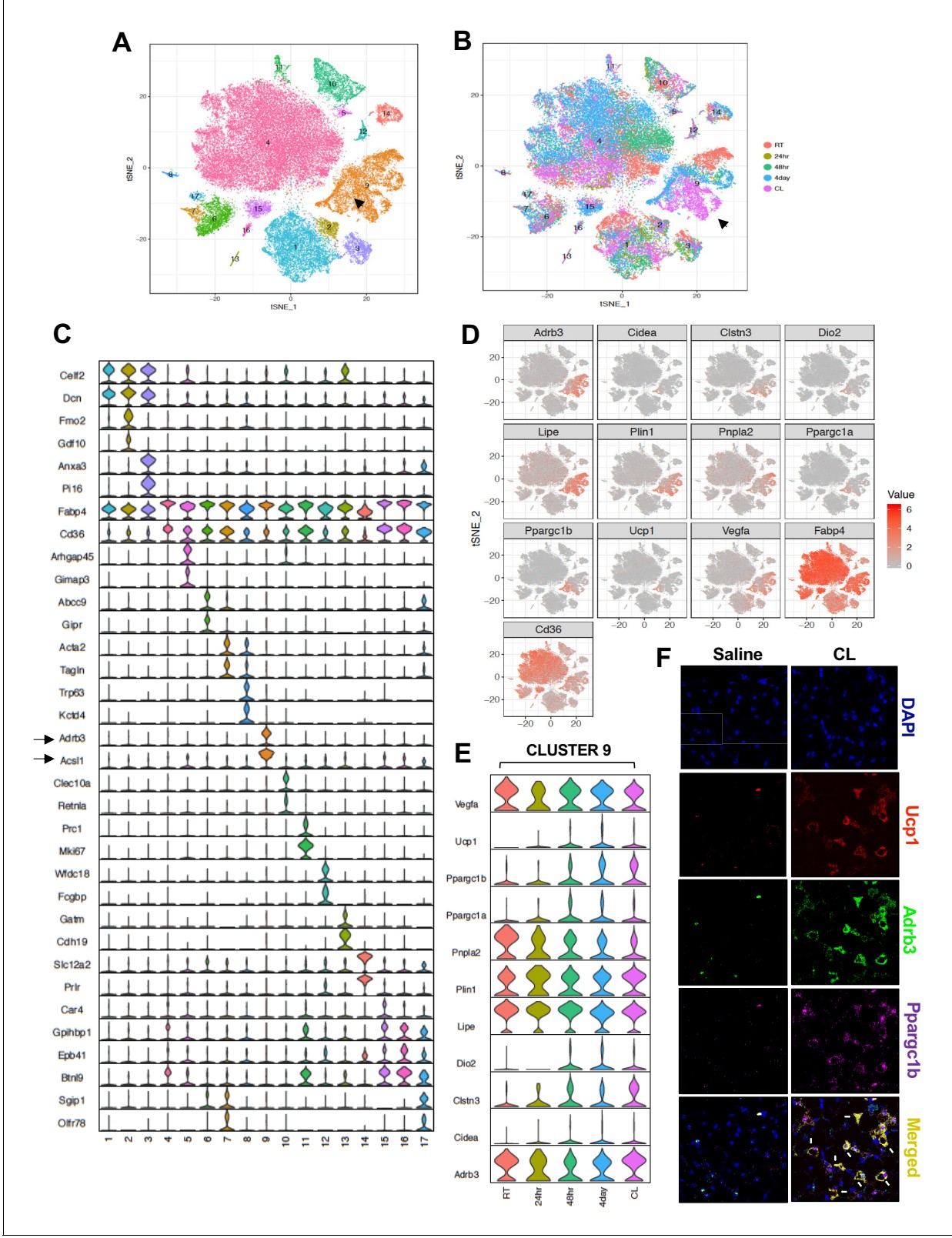

**Figure 4.** Transcriptomic visualization of thermogenic adipocytes in iWAT. (**A**) Aggregated tSNE-plot showing 17 clusters from ~54,000 adipocytes derived from iWAT of mice at RT, cold-exposed (24 hr, 48 hr, and 4 days), or treated with CL for 4 days at 1 mg/kg/day. Each colored dot is an adipocyte that falls into a cluster based on transcriptomic signature. (**B**) tSNE-plot showing indicated treatment-dependent sub-clustering of aggregated clusters shown in (**A**). Black arrow indicates the Type nine adipocyte cluster. (**C**) Normalized expression values of the top two adipocyte

*Figure 4 continued on next page*

*Figure 4 continued*

subtype-specific cluster genes from (**A and B**) plotted as violin plots with clusters as rows and genes as columns. (**D**) tSNE-plot showing distribution of indicated genes from adipocytes from (**A**). (**E**) Normalized expression values of indicated genes in the Type nine adipocyte cluster under different treatment condition plotted as violin plots with treatment conditions as rows and genes as columns. (**F**) RNAScope FISH (see Materials and methods) of indicated probes from iWAT of mice treated with saline or CL for 4 days at 1 mg/kg/day.
DOI: https://doi.org/10.7554/eLife.49501.007

The following figure supplements are available for figure 4:

**Figure supplement 1.** Enrichment of cells involved in lipid mobilization in Cluster nine adipocytes.
DOI: https://doi.org/10.7554/eLife.49501.008

**Figure supplement 2.** Type nine cluster constitute of thermogenic adipocytes.
DOI: https://doi.org/10.7554/eLife.49501.009

presented above, adipocytes from AdIL10Rα KO mice showed a specific increase in the population of thermogenic adipocytes (cluster 9) under both basal conditions (RT) and in response to thermogenic stimuli (cold exposure or CL treatment) (*Figure 5B*). The dot-plot in *Figure 5C* shows that the overall pattern of adipocyte cluster enrichment was similar to that in *Figure 5A*-bottom, and only type nine adipocytes showed a positive shift in cell fraction upon IL10R depletion, except for 48 hr cold exposure where we observed unexpected 4% decrease in cell fraction.

Type nine adipocytes from AdIL10RαKO mice were more metabolically active and showed increased expression of genes linked to mitochondrial activity, energy derivation from FFA, and positive regulation of cold-induced thermogenesis (*Figure 5—figure supplement 1A*). A Volcano plot of the data revealed that genes involved in lipid mobilization and adipose thermogenesis (such as *Lipe, Nr1d1, Oplah, Nfkbia, Pck1, Cebpb, Vegfa, Angptl4*) were increased, and genes correlated with obesity and adiposity such as (*Nrip1, Lpl, Zbtb20, Acss2*) were decreased in iWAT adipocytes from AdIL10RαKO mice compared to controls (*Figure 5D*). However, not all adipogenic and thermogenic genes were different between AdIL10Rα KO and control adipocytes. For example, pan-adipocyte genes such as *Cd36, Fabp4*, and *Aqp1* were similarly expressed between cells of both genotypes (*Figure 5—figure supplement 1B*). Overall, our SNAP-seq data reveal previously unappreciated heterogeneity of mature adipocytes in subcutaneous adipose tissue and point to the existence of distinct cell populations with potentially specialized biological functions. These data further show that IL10 signaling in fat tissue targets a distinct, highly metabolically active and thermogenic adipocyte population.

## scRNA-Seq of iWAT stromal vascular fraction reveals a role for IL10-expressing adaptive immune cells in regulation of adipose thermogenesis

Prior bone marrow transplantation experiments had shown that IL10 produced by one or more hematopoietic cell types could rescue the thermogenic phenotype of global IL10-deficient mice (*Rajbhandari et al., 2018*). Therefore, we examined changes in non-adipocyte cells types within iWAT in the setting of thermogenic stimuli to evaluate their potential contribution to IL10/IL10R signaling. We treated WT mice with saline or β3-adrenergic agonist (CL; 1 mg/kg/day) for 4 days, separated iWAT stromal cells from adipocytes, and performed scRNA-Seq on ~10,000 isolated stromal vascular cells (SVF) per mouse as described in Materials and methods. t-SNE plotting of the data revealed 12 major cell clusters (*Figure 6A*). Further subclustering analysis based on known cell marker genes identified four clusters of adipocyte precursor cells (APCs), four clusters of B cells, three clusters of macrophages, and four clusters of T cells (*Figure 6A*).

To gain insight into the remodeling of stromal cells under adrenergic stress, we segregated the cumulative tSNE-plot into CL and saline treatment. The tSNE and dot plots in *Figure 6B and C* reveal a relative depletion of APCs and a major increase in B-cell populations in iWAT upon CL treatment. Since adaptive immune cells (T- and B-cells) are major potential producers of IL10 (*Saraiva and O'Garra, 2010*), we further examined IL10 transcript levels in our data set. t-SNE-plotting showed that total IL10 transcripts increased ~3 fold upon CL treatment in B-cell clusters (*Figure 6D*). To quantitatively determine IL10 transcript levels in saline and CL- treated conditions, we plotted fold change of the ratio of CL and saline as a function of p value. Compared to saline (represented by dotted line) *Il10* was markedly upregulated in T- and B-cell populations (*Figure 6E*).

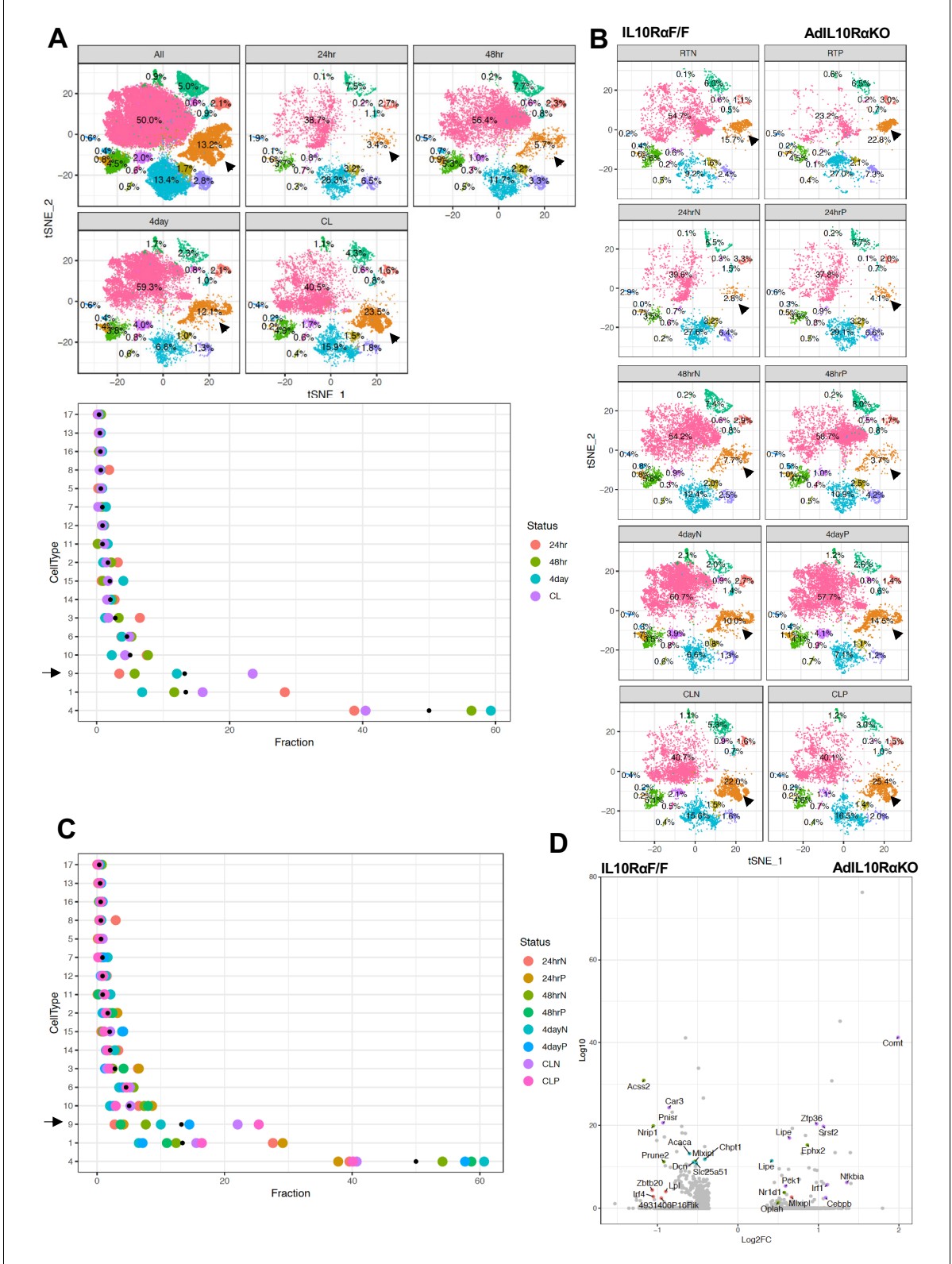

**Figure 5.** Enrichment of the thermogenic adipocyte cluster in IL10Rα-depleted mice. (A) Top: tSNE-plot showing percentage of clusters from aggregated (All) or from indicated treatments. Bottom: Colored dot plot showing percentage of fractions plotted in y-axis and cell types in x-axis under indicated conditions. (B) tSNE-plot showing differences in percentage of clusters between adipocytes from iWAT from control and AdIL10Rα KO mice under RT, 24 hr, 48 hr or 4 days cold exposure, or CL treated conditions. (C) Colored dot plot showing percentage of fractions plotted in y-axis and cell

*Figure 5 continued on next page*

Figure 5 continued

types in x-axis under indicated conditions and genotype. N deonotes Adipoq-Cre-Negative IL10RαF/F (IL10RαF/F) and P denotes Adipoq-Cre-Positive (AdIL10Rα KO) mice. (D) Volcano plot showing adipocyte gene expression differences between indicated mice from cluster 9. The Log2Fold Change (FC) ratio of floxed control vs. AdIL10Rα KO was plotted as a function of log10 p-value, with select genes indicated with text. Black arrow indicates the Type nine adipocyte cluster.

DOI: https://doi.org/10.7554/eLife.49501.010

The following figure supplement is available for figure 5:

**Figure supplement 1.** Thermogenic pathway is enriched in the Type 9 cluster of IL10Rα-deficient adipocytes.

DOI: https://doi.org/10.7554/eLife.49501.011

To test the possibility that production of IL10 by iWAT-resident T and B cells might contribute to the regulation of thermogenesis, we treated WT or functional T- and B-cell–deficient SCID mice (*Bosma et al., 1988*) with CL or exposed them to 4˚C for 24 hr. In agreement with our hypothesis, the thermogenic gene program was enhanced in the iWAT of SCID mice compared to controls (*Figure 7A,B*). Accordingly, SCID mice also had higher EE and oxygen consumption and decreased RER, as measured by metabolic chamber studies. (*Figure 7C and D*). Collectively, these results suggest that lymphocytes are an important source of the IL10 acting on adipocyte IL10 receptors during thermogenesis.

## Discussion

The influence of inflammation on obesity and adipose insulin resistance has been studied extensively; however, the role of adipose-resident immune cells in regulating the balance between adiposity, lipid mobilization, and thermogenesis is incompletely understood. We previously reported that hematopoietic-secreted IL10 inhibits adrenergic signaling-mediated lipid mobilization and thermogenesis (*Rajbhandari et al., 2018*). Here we provide evidence from scRNA-seq analysis of both adipose stromal-vascular cells and mature adipocytes indicating that adrenergic stimulation causes an increase in the abundance of IL10-secreting adaptive immune cells, and that this cytokine acts directly on the IL10Rα complex in mature adipocytes to antagonize thermogenesis. Genetic ablation of IL10Rα in adipocytes increases the browning of white adipose tissue and selectively enhances thermogenic gene expression, defining mature adipocytes as the primary target of the metabolic actions of IL10. Our SNAP-seq data further revealed that adipose tissue is composed of surprisingly complex subpopulation of adipocytes, including distinct subtypes whose gene expression suggests they are subspecialized for different processes, including lipogenesis and thermogenesis. Deletion of IL10Rα selectively increases the subpopulation of metabolically active, thermogenic adipocytes.

The immune system plays an important role in maintaining adipose homeostasis. Prior studies have shown that signaling from both innate and adaptive immune cells influences lipid handling, adipocyte size and function, and whole body lipid homeostasis (*Bapat et al., 2015*; *Schäffler and Schölmerich, 2010*; *Sell et al., 2012*; *Wernstedt Asterholm et al., 2014*). In a proinflammatory state, as seen in adrenergic stress, cancer cachexia, and burn victims, adipose tissue undergoes remodeling that activates lipid mobilization and thermogenesis which can lead to lipodystrophy (*Patsouris et al., 2015*; *Petruzzelli et al., 2014*). One established mechanism to block uncontrolled lipolysis is through catabolism of catecholamines. Sympathetic-associated macrophages, NRLP3 inflammasomes, and OCT3 have all been shown to enhance catecholamine clearance in adipose tissue (*Camell et al., 2017*; *Pirzgalska et al., 2017*; *Song et al., 2019*). Immune cells also could counteract adrenergic signaling by releasing factors that prevent excessive lipolysis and thermogenesis and thereby direct adipocytes to reserve energy in the setting of starvation or infection. This idea is supported by our scRNA-seq studies on iWAT SVFs of mice under adrenergic stress. Mice treated with β3-adrenergic agonist have expanded adaptive immune cell populations and a 3-fold increase the abundance of IL10 transcript in adipose SVF. Increased production of IL10 in the microenvironment could antagonize adrenergic-mediated thermogenesis, potentially providing a mechanism whereby immune cells help to maintain lipid homeostasis in the setting of energy demand.

A recent study has shown that deletion of Oct coactivator from B-cells (OcaB), which is essential for B-lymphocyte maturation and development, promotes adipose browning and protects mice from age-induced insulin resistance (*Carter et al., 2018*). We showed here that the absence of IL10-

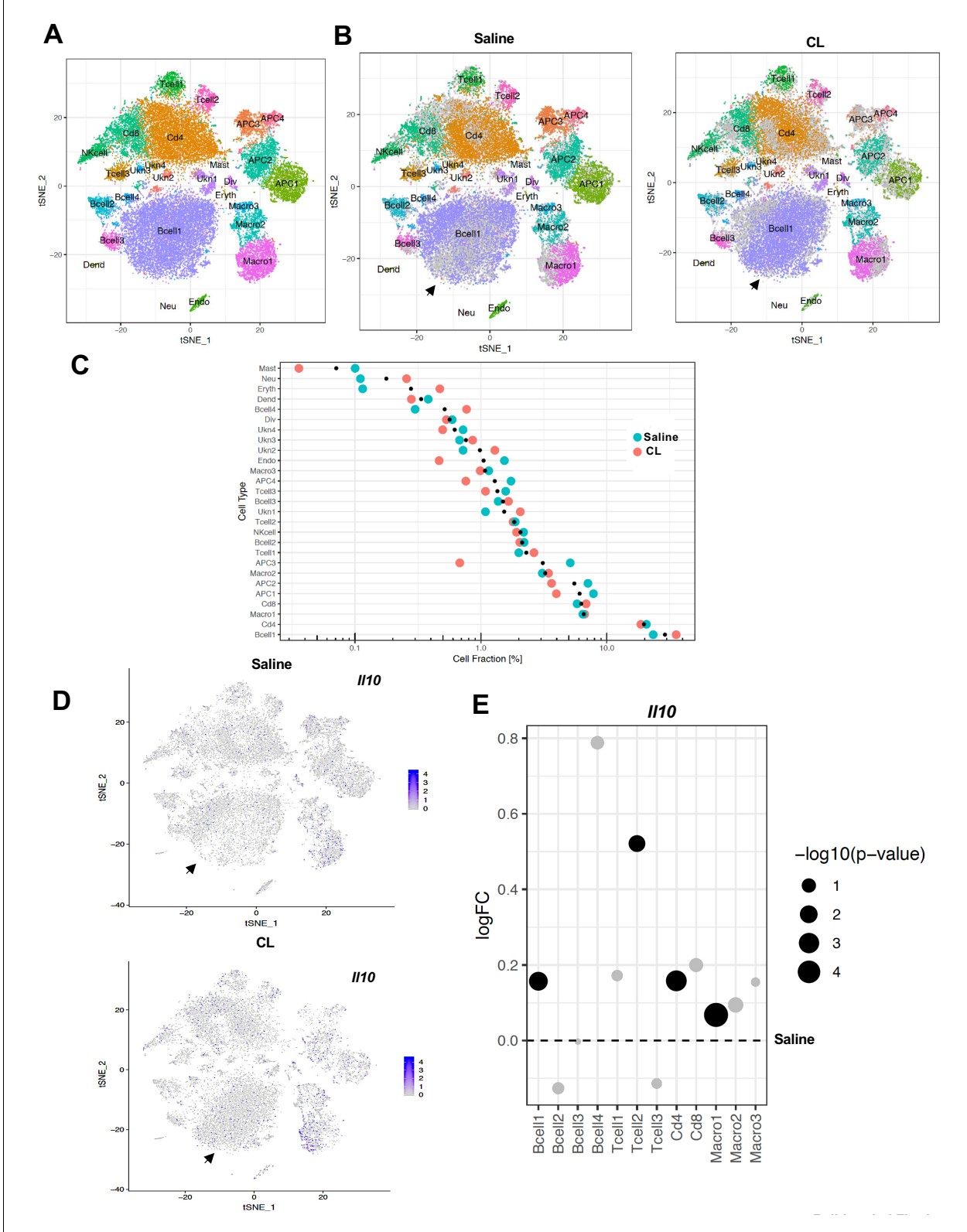

**Figure 6.** scRNA-Seq reveals an increase in adaptive immune cells under adrenergic stress. (**A**) Aggregated tSNE-plot of combined ~10,000 SVF cells isolated from iWAT of mice treated with CL or saline for 4 days. Colored dots are cells assigned to clusters based on similar transcriptomic signatures and these clusters correspond to specific cell-types. (**B**) Segregated tSNE-plot from (**A**) showing percentage of cell-types between control (Saline) and β3-agonist CL-treated mice. The tSNE-plot also shows differences in percentage of clusters between control and CL-treated mice. Black arrows indicate

*Figure 6 continued on next page*

*Figure 6 continued*
major B-cell population. (**C**) Colored dot plot showing percentage of fractions plotted in y-axis and cell types in x-axis under indicated conditions. (**D**) tSNE-plot showing cells expressing *Il10* in control and CL-treated mice. Black arrows indicate major cell clusters with *Il10* expression. (**E**) Dot plot showing expression levels of *Il10* in indicated cells comparing control and CL-treated mice. The Log2Fold Change (FC) ratio of saline vs. CL was plotted as a function of log10 p-value and indicated as different sizes of dots. Fold-change and p-value was compared with saline condition represents as a dotted line.
DOI: https://doi.org/10.7554/eLife.49501.012

secreting adaptive immune cells in mice also leads to enhanced adipose thermogenesis, a finding consistent with the phenotypes of both IL10 KO and AdIL10Rα KO mice.

To more deeply interrogate the effect of IL10 signaling on adipocyte identity and function, we performed SNAP-seq of adipocytes from iWAT. Although prior studies have used FACS and immortalized clonal preadipocytes to assess different types of adipocytes in various adipose tissue depots (*Hagberg et al., 2018*; *Lee et al., 2019*), to our knowledge, this is the first report of rigorous single nuclei RNA-Seq of adipocytes in the setting of a thermogenic challenge. Our data revealed distinct clusters reflecting subsets of mature adipocytes with differential gene expression. Our data show that the mature tissue adipocyte population is much more heterogeneous than previously appreciated. We categorized by 14 distinct subsets of mature adipocytes that appear to be specialized to participate in at least partially distinct metabolic pathways. These findings suggest that different functions of adipose tissue may be executed by different cell populations, rather than similar cell populations performing diverse functions. Further studies will be needed to test this idea.

Interestingly, Type nine adipocytes were highly metabolically active and thermogenic, and we speculate that this cluster corresponds to the so-called 'beige' adipocyte population. Violin plots of the Type nine cluster showed selective expression of genes involved in lipid mobilization, such as *Adrb3, Acsl2, Lipe, Pnpla2*, and brown/beige-associated genes such as *Ucp1, Cidea, Dio2*, and *Clstn3*. The identity of this cluster was further confirmed by performing snRNA-seq on adipocytes from mice at RT, exposed to cold for various times, or treated with CL. We saw a gradual increase in the percentage of Type nine adipocytes with increasing thermogenic stimulus.

Interestingly, we also noticed distinct sub-clustering of Type nine adipocytes within the cluster. Cells expressing genes involved in lipolysis had an even distribution, whereas cells with higher expression of brown/beige genes were subclustered together after 4 days of cold or CL treatment. This data suggests that *Adrb3*-expressing adipocytes have varying browning capacity. We also noticed that the subclustering patterns of cold-exposed versus CL-treated Type nine adipocytes were also different. This likely reflects different transcriptional responses to cold exposure and pharmacological adrenergic receptor activation. DEG analysis of AdIL10Rα KO and control adipocytes showed that ablation of IL10Rα leads to differential enrichment of adipocyte populations. Type nine adipocytes appear to be enriched in genes that are highly expressed in AdIL10Rα KO compared to control iWAT. Furthermore, AdIL10Rα KO adipocytes are more metabolically active and have heightened response to adrenergic stimulation compared to controls. These data suggest that deletion of IL10Rα from adipocytes leads to the selective enrichment of metabolically active adipocytes that could ultimately lead to increased response to adrenergic signaling and enhanced thermogenesis.

In conclusion, these data provide insight into crosstalk between IL-10-secreting immune cells and adipocytes within adipose tissue, as well as into the complexity of the transcriptional response to adrenergic signaling in mature adipocytes. A better understanding of the pathways influencing the development and phenotypic transformation of Type nine adipocytes could ultimately lead to strategies to increase energy expenditure and protect against diet-induced obesity.

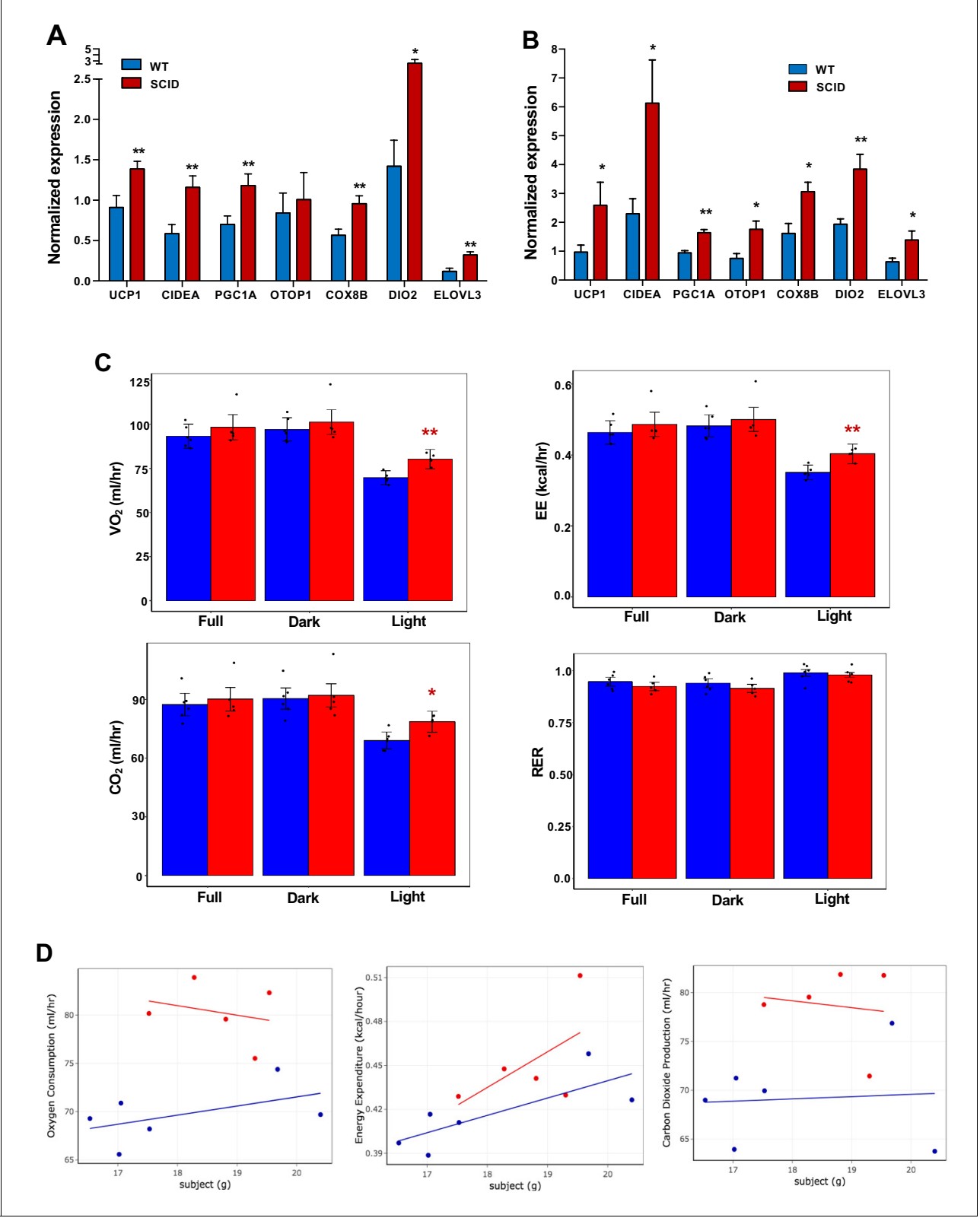

**Figure 7.** Increased adipose thermogenesis and EE in lymphocyte-deficient mice. (**A and B**) Real-time PCR analysis of gene expression in iWAT from chow-fed 10 week 24 hr cold-exposed (**A**) or CL (1 mg/kg/day for 4 days (**B**) mice. (**C, D**) Bar graphs (C) and scatter plot (D) (Light cycle) showing oxygen consumption, energy expenditure, carbon dioxide production, and RER in CL-treated mice with body mass as covariate. N = 6,5. *, p<0.05, **, p<0.01.

DOI: https://doi.org/10.7554/eLife.49501.013

# Materials and methods

## Key resources table

| Reagent type (species) or resource | Designation | Source or reference | Identifiers | Additional information |
|---|---|---|---|---|
| Genetic reagent sample (*M. musculus*) | *Il10ra*$^{flox}$ | Jackson Laboratory | Stock No. 028146 RRID:MGI:189797 | PMID: 22711892 |
| Genetic reagent sample (*M. musculus*) | C57BL/6J | Jackson Laboratory | Stock No. 000664 | |
| Genetic reagent sample (*M. musculus*) | C57BL/6J SCID | Jackson Laboratory | Stock No. 001913 RRID:MGI:14443 | PMID: 8339285 |
| Genetic reagent sample (*M. musculus*) | Adipoq-Cre | Jackson Laboratory | Stock No. 010803 RRID:MGI:168971 | PMID: 21356515 |
| Chemical compound | β3-adrenergic receptor agonist (CL 316, 243; CL) | Sigma | C5976 | 1 mg/kg/day |
| Chemical compound | Collagenase II | Worthington Biochemical | LS004176 | 3 mg/ml |
| Chemical compound | Collagenase D | Sigma | 11088882001 | 9.8 mg/ml |
| Commercial kit | EvaGreen droplet generation oil | BIO-RAD | 1864005 | |
| Commercial kit | ChemGenes barcoded microparticles | ChemGenes | N/A | |
| Commercial kit | FlowJEM aquapel-treated microfluidic device | FlowJEM | N/A | |
| Commercial kit | Nextera DNA Library Preparation kit | Illumina | 2001874 | |
| Commercial kit | Minute nuclei and cytosol isolation kit for Adipose | Invent Biotech | AN-029 | |
| Commercial kit | 40 μm cell strainer | Bel-Art FlowMI | H13680-0040 | |
| Commercial kit | Single Cell 3' Chip | 10X Genomics | 1000127 | |
| Commercial kit | Chromium Single Cell 3' library and Cell Bead kit | 10X Genomics | 1000128 | |
| Commercial kit | TruSeq Stranded Total RNA Library Prep Kit | Illumina | RS-122–2102 | |
| Commercial kit | RNAscope Multiplex Flurorescent Reagent kit v2 | Advanced Cell Diagnostics | 323316 | |
| Commercial kit | Opal 520 | Akoya Biosciences | FP1487001KT | 1:1000 |
| Commercial kit | Opal 570 | Akoya Biosciences | FP1488001KT | 1:1000 |
| Commercial kit | Opal 620 | Akoya Biosciences | FP1495001KT | 1:1000 |
| Commercial kit | mm-Adrb3 | Advanced Cell Diagnostics | 495521 | |
| Commercial kit | mm-UCP1 | Advanced Cell Diagnostics | 455411 | |
| Commercial kit | mm-Ppargc1b | Advanced Cell Diagnostics | 402131 | |
| Commercial kit | TRIzol | Thermo Fischer | 15596026 | |

*Continued on next page*

*Continued*

| Reagent type (species) or resource | Designation | Source or reference | Identifiers | Additional information |
|---|---|---|---|---|
| Commercial kit | iScript cDNA synthesis kit | Bio-Rad | 1708890 | |
| Commercial kit | SYBER Green Master mix | Diagenode | DMSG-2X-A300 | |
| Commercial kit | RIPA Lysis buffer | Boston Bioproducts | BP-115 | |
| Antibody | IL10Rα | R and D System | AF-474-SP | |
| Antibody | α-Tubulin | Calbiochem | CP06 RRID:AB_2617116 | |
| Antibody | Actin | Sigma | A2066 RRID:AB_476693 | |
| Software | Drop-seq tools version 1.13 | https://github.com/broadinstitute/Drop-seq | | |
| Software | dropSeqPipe | https://github.com/darneson/dropSeqPipeDropEST | | |
| Software | Mouse Genome Allignment mm10 | STAR-2.5.0c | | https://github.com/alexdobin/STAR |
| Software | Seurat R package version 3.0.0.9000 | https://github.com/satijalab/seurat | | |
| Software | Cell Ranger V3.0.2 | 10X Genomics | | |
| Other | Prolong Diamond Antifade Mountant with DAPI | Thermo Fischer | P36966 | |
| Other | HFD; 60% kcal fat | Research Diets | D12492 | |

## Animal studies

Breeding pairs of *Il10RaF/F*mice (#028146), Adiponectin CRE (#010803), C57BL/6 SCID (#001913) and C57BL/6 WT controls (#000664) were acquired from Jackson Laboratory and maintained in a pathogen-free barrier-protected environment (12:12 hr light/dark cycle, 22–24°C) at the UCLA animal facility. Experimental mice were sacrificed at ages mentioned in figure legends for gene expression analysis. For the time course cold exposure experiment, WT mice at 8–10 weeks of age were singly housed at 4°C room in a non-bedded cage without food and water for first 6 hr; thereafter food, water, and one cotton square were added. For the 24 hr harvest, 3 hr before harvest, food, water, and cotton square were removed and then mice were harvested. For the 48 hr and 4 day cold exposure, cages were changed daily with new cotton squares and 3 hr before the time of harvest, the food, water, and cotton square were removed. For β-adrenergic stimulation experiments, mice were intraperitoneally injected with β3-adrenergic agonist (CL 316,243; CL, at 1 mg/kg/day for 4 days) or saline. For the diet study, 10-week-old *Il10ra^{F/F}* and AdIL10RαKO mice were fed a 60% high-fat diet (Research Diets) for the indicated times. Mouse weights were measured every week and food was replaced weekly. At the end of the experiment, iWATs were resected for gene expression analysis. Indirect calorimetry was performed using a Columbus Instruments Comprehensive Lab Animal Monitoring System (CLAMS, Columbus Instruments). Animals were placed individually in chambers for three consecutive days at ambient temperature (26.5°C) with 12 hr light/dark cycles. Animals had free access to food and water. Respiratory measurements were made in 20 min intervals after initial 7–9 hr acclimation period. Energy expenditure was calculated from VO2 and RER using the Lusk equation, EE in Kcal/hr = (3.815 + 1.232 X RER) X VO2 in ml/min. CLAMS data were analyzed by CALR web-based software (*Mina et al., 2018*). Animal experiments were conducted in accordance with the UCLA Institutional Animal Care and Research Advisory Committee.

## RNA-Seq

Total RNA was prepared as described (*Tong et al., 2016*). Strand-specific libraries were generated from 500 ng total RNA using the TruSeq Stranded Total RNA Library Prep Kit (Illumina). cDNA libraries were single-end sequenced (50 bp) on an Illumina HiSeq 2000 or 4000. Reads were aligned to the mouse genome (NCBI37/mm9) with TopHat v1.3.3 and allowed one alignment with up to two mismatches per read. mRNA RPKM values were calculated using Seqmonk's mRNA quantitation pipeline. A gene was included in the analysis if it met all the following criteria: the maximum RPKM reached four at any time point, the gene length was >200 bp, and for in-vitro studies was induced at least 3-fold from Day 0 samples, and the expression was significantly different from the basal ($p<0.01$) as determined by the DESeq2 package in R Bioconductor. P-values were adjusted using the Benjamini-Hochberg procedure of multiple hypothesis testing (*Benjamini and Hochberg, 1995*).

## scRNA-Seq of adipose stromal vascular fraction (SVF) population

### Single cell isolation from SVF

Inguinal white adipose tissue (iWAT) from mice treated with saline or CL were dissected and placed on sterile 6-well tissue culture plate with ice-cold 1X DPBS. Fat pads were blotted on a napkin to removed excess liquid. Tissues were cut and minced with scissors and placed in 15 ml conical tubes containing digestion buffer (2 ml DPBS and Collagenase II at 3 mg/ml; Worthington Biochemical, Lakewood, NJ, USA) and incubated at 37°C for 40 min with gentle shaking at 100 rpm. Following tissue digestion 8 ml of resuspension media (DMEM/F12 with glutamax supplemented with 15%FBS and 1% pen/strep; Thermo Scientific, CA) was added to stop enzyme activity. The digestion mixture was passed through 100 µm cell strainer and centrifuged at 150 x g for 8 min at room temperature. The pellet was resuspended and incubated in RBC lysis buffer (Thermo Scientific, CA) for 3 min at room temperature to remove red blood cells followed by centrifugation at 150 x g for 8 min. The pellet was resuspened in resuspension media and spun down again at 150 x g for 8 min. Finally, the cell pellet was resuspended in 1 ml of 0.01% BSA (in DPBS). This final cell suspension solution was passed through a 40 µm cell strainer (Fisher Scientific, Hampton, NH, USA) to discard debris and cell number was counted for Drop-Seq application.

## Drop-seq single cell barcoding and library preparation

The Drop-seq protocol from Macosko et al. and version 3.1 of the online Drop-seq protocol [http://mccarrolllab.com/download/905/] was followed with minor modifications to generate STAMPs (single-cell transcriptomes attached to microparticles) and cDNA libraries (*Macosko et al., 2015*). Briefly, to create oil droplets with barcoded cells, single cell suspensions (100 cells/µl), EvaGreen droplet generation oil (BIO-RAD, Hercules, CA, USA), and ChemGenes barcoded microparticles (ChemGenes, Wilmington, MA, USA) were co-flowed through a FlowJEM aquapel-treated microfluidic device (FlowJEM, Toronto, Canada) at the recommended flow speeds (oil: 15,000 µl/hr, cells: 4000 µl/hr, and beads 4000 µl/hr). After STAMP generation, oil droplets were broken, and cDNA synthesis was performed. To obtain enough cDNA for library preparations the Drop-seq protocol was followed with the following modifications. For the PCR step, 4000 beads were used per tube, the number of cycles was changed to 4 + 11 and multiple PCR tubes were pooled. cDNA library concentration and quality were assessed using the Agilent TapeStation system (Agilent, Santa Clara, CA, USA). The samples were then tagmented using the Nextera DNA Library Preparation kit (Illumina, San Diego, CA, USA) and multiplex indices were added. After another round of PCR, the samples were assessed on a Tapestation high sensitivity DNA screentape (Agilent, Santa Clara, CA, USA) for library quality before sequencing.

## Illumina high-throughput sequencing of Drop-seq libraries

Molar concentrations of the Drop-seq libraries were quantified using the Qubit Fluorometric Quantitation system (ThermoFisher, Canoga Park, CA, USA) and library fragment lengths were estimated using a Tapestation high sensitivity DNA screentape (Agilent, Santa Clara, CA, USA). Samples were normalized by concentration and then pooled appropriately. Pooled libraries were then sequenced on an Illumina HiSeq 4000 (Illumina, San Diego, CA, USA) instrument using the Drop-seq custom read 1B primer (GCCTGTCCGCGGAAGCAGTGGTATCAACGCAGAGTAC) (IDT, Coralville, IA, USA)

and PE100 reads were generated. Read 1 consists of the 12 bp cell barcode, followed by the 8 bp UMI, and the last 80 bp on the read are not used. Read two contains the single cell transcripts.

## Drop-seq data pre-processing and quality control

Demultiplexed fastq files generated from Drop-seq were processed to digital expression gene matrices (DGEs) using Drop-seq tools version 1.13 (https://github.com/broadinstitute/Drop-seq) and dropEst (*Petukhov et al., 2018*). The workflow is available as modified version of the snakemake-based dropSeqPipe (https://github.com/Hoohm/dropSeqPipe) workflow and is available on github (*Arneson, 2019*; copy archived at https://github.com/elifesciences-publications/dropSeqPipeDrop-EST). Briefly, fastq files were converted to BAM format and cell and molecular barcodes were tagged. Reads corresponding to low quality barcodes were removed and any occurrence of the SMART adapter sequence or polyA tails found in the reads was trimmed. These cleaned reads were converted back to fastq format to be aligned to the mouse reference genome mm10 using STAR-2.5.0c. After the reads were aligned, the reads which overlapped with exons, introns, and intergenic regions were tagged using a RefFlat annotation file of mm10. To make use of reads aligning to intronic regions, which are not considered in Drop-seq tools v1.13, we used dropEst to construct digital gene expression matrices from the tagged, aligned reads where each row in the matrix is the read count of a gene and each column is a unique single cell. The count values for each cell were normalized by the total number of UMIs in that cell and then multiplied by 10,000 and log transformed. Single cells were identified from background ambient mRNA using thresholds of at least 700 transcripts and a maximum mitochondrial fraction of 10%.

## Identification of cell clusters

The Seurat R package version 3.0.0.9000 (https://github.com/satijalab/seurat) was used to project all sequenced cells onto two dimensions using t-SNE and Louvain (*Blondel et al., 2008*; *van der Maaten and Hinton, 2008*) clustering was used to assign clusters. The optimal number of PCs used for t-SNE/UMAP dimensionality reduction and Louvain clustering was determined using the Jackstraw permutation approach and a grid-search of the parameters. Similarly, the density used to assign clusters was identified using a parameter grid search.

## Identification of marker genes of individual cell clusters

We defined cell cluster specific marker genes from our Drop-seq dataset using the FindConservedMarkers function in Seurat across all the samples. Briefly, a Wilcoxon Rank Sum Test is run within each sample and a meta p-value across all samples is computed to assess the significance of each gene as a marker for a cluster. Within each sample, the cells are split into two groups: single cells from the cell type of interest and all other single cells. To be considered in the analysis, the gene had to be expressed in at least 10% of the single cells form one of the groups and there had to be at least a 0.25 log fold change in gene expression between the groups. This process was conducted within each sample separately, and then a meta p-value was assessed from the p-values across all samples. Multiple testing was corrected using the Benjamini-Hochberg method on the meta p-values and genes with an FDR < 0.05 were defined as cell type specific marker genes.

## Resolving cell identities of the cell clusters

 We used two methods to resolve the identities of the cell clusters. First, we used known cell-type specific markers curated from literature, single cell atlases, previous studies in the SVF and PBMCs, and from Immgen (immgen.org) to find distinct expression patterns in the cell clusters (*Burl et al., 2018*; *Chen et al., 2018*; *Han et al., 2018*; *Hepler et al., 2018*; *Tabula Muris Consortium et al., 2018*; *Stoeckius et al., 2017*; *Zhang et al., 2019*). A cluster showing unique expression of a known marker gene can be used to identify that cell type. To consider more than a single gene, we evaluated the overlap between known cell type marker genes with the marker genes identified in our cell clusters using FindConservedMarkers. Significant overlap was assessed using a Fisher's exact test with Bonferroni correction for multiple testing. The two methods showed consistency in cell identity determination. The GEO accession number for the sequencing data is GSE133486.

## Single nuclei adipocyte RNA-Sequencing (SNAP-Seq)

### Adipocyte nuclei isolation from iWAT

200–400 mg of inguinal white adipose tissues (iWAT) from mice exposed to conditions mentioned in the text were placed on sterile 6-well tissue culture plate with ice-cold 1XPBS. Fat pads were blotted on a napkin to removed excess liquid. Tissues were cut and minced with scissors and were placed in 15 ml conical tubes containing digestion buffer (DPBS and Collagenase D at 9.8 mg/ml; Sigma, MO) at incubated at 37°C for 45 mins with gentle shaking at 100 rpm. 10 ml of resuspension media (DMEM/F12 with glutamax supplemented with 15% FBS and 1% pen/strep; Thermo Scientific, CA) was added to digested solution and slowly inverted five times. The digestion mixture was centrifuged at 200 x g for 5 mins at RT. Floating adipocytes were collected using P1000 pipet with half cut P1000 tip. Adipocytes were transferred to a new 15 ml tube and kept on ice for five mins. Excess liquid was aspirated using 1 ml syringe and adipocytes were then washed with 1 ml DPBS and the suspension was spun down at 200 g for 5 mins at RT. Spun down liquid was aspirated using 1 ml syringe and adipocyte nuclei were isolated using Minute nuclei and cytosol isolation kit for adipose tissue using manufacture's instruction (Invent Biotechnologies, MN) with modifications. Briefly, adipocytes were slowly resuspended in 600 µl nuclei lysis buffer (N/C Buffer) and lysate was transferred to a filter cartridge with collection tube and incubated at −20°C freezer for 20 min with cap open. After incubation, the tube was centrifuged at 2000 rpm for 2 min at 4°C. The filter cartridge was discarded without agitation and the collection tube was immediately centrifuged at 4000 rpm for 4 min at 4°C. Supernatant was gently removed using P200 pipet without touching the side walls. Nuclei were resuspended in 30 ul of nuclei resuspension buffer (DPBS+0.1%BSA) per 200–400 mg of iWAT (i.e. one 8–10 week chow fed mouse). For SNAP-seq, 2–3 mice were combined and 60 µl of nuclei suspension was transferred to a new 2 ml tube and resuspended with 500–700 µl of nuclei resuspension buffer and filtered using 40 µm cell strainer (Flowmi Cell Strainer, Belart, NJ) twice to get clean single nuclei suspension. As shown in *Figure 3A*, for quality control, nuclei were first DAPI stained and then filtered or FACS sorted to get single nuclei suspension. After microfluidic partitioning in 10xGenomics platform (see below), nuclei lysis was checked by observing oil emulsion under fluorescent microscope for DAPI diffusion.

### Adipocyte single nuclei barcoding and library preparation

Approximately 10,000 nuclei were loaded onto Single Cell 3' Chip (10xGenomics, CA) per channel with an expected recovery to 4000–7000 nuclei. The Chip was placed on a 10XGenomics Instrument to generate single nuclei gel beads in emulsion (GEMs). For optimal nuclei lysis, GEMs were incubated on ice for 50 mins. After incubation, single nuclei RNA-Seq libraries were prepared using Chromium Single Cell 3' Library and Cell Bead Kit) according to manufacturer's instruction.

### Illumina high-throughput sequencing libraries

The 10X genomics library molar concentration was quantified by Qubit Fluorometric Quantitation (ThermoFisher, Canoga Park, CA, USA) and library fragment length was estimated using a TapeStation (Aligent, Santa Clara, CA, USA). Sequencing was performed on an Illumina HiSeq 4000 (Illumina, San Diego, CA, USA) instrument with PE100 reads and an 8 bp index read to multiplexing. With the version three chemistry, the first 26 bp of Read 1 consist of the cell barcode and the UMI, and the last 74 bp on the read are not used. Read two contains the single cell transcripts.

### SNAP-Seq data pre-processing and quality control-

Digital gene expression matrices (DGEs) in sparse matrix representation we obtained using 10x Genomics' Cell Ranger v3.0.2 software suite. Briefly,. bcl files obtained from the UCLA Broad Stem Cell Research Center sequencing core were demultiplexed and converted to fastq files using the mkfastq function in Cell Ranger which wraps Illumina's bcl2fastq v2.19.1.403. The counts function in Cell Ranger was used to generate DGEs from the fastq files. Briefly, the resulting fastq files are aligned to a 10x supplied mm10 reference genome (mm10-3.0.0) using STAR and reads are identified as either exonic, intronic, or intergenic using the supplied 10x Genomics GTF file. To determine cell barcodes, the counts function in Cell Ranger implements an algorithm based on EmptyDrops (*Lun et al., 2019*). Only reads which align to exonic regions were used in the resulting DGE. The count values for each cell were normalized by the total number of UMIs in that cell and then

multiplied by 10,000 and log transformed. Single cells were identified from background ambient mRNA using thresholds of at least 200 genes and a maximum mitochondrial fraction of 10%.

## Identification of adipocyte clusters

The Seurat R package version 3.0.0.9000 (https://github.com/satijalab/seurat) was used to project all sequenced cells onto two dimensions using t-SNE/UMAP and Louvain clustering was used to assign clusters. The optimal number of PCs used for t-SNE dimensionality reduction and Louvain clustering were determined using the Jackstraw permutation approach and a grid-search of the parameters. Similarly, the density used to assign clusters was identified using a parameter grid search.

## Identification of marker genes of individual adipocyte clusters

We defined cell cluster specific marker genes from our 10x Genomics dataset using the FindConservedMarkers function in Seurat across all the samples. Briefly, a Wilcoxon Rank Sum Test is run within each sample and a meta p-value across all samples is computed to assess the significance of each gene as a marker for a particular cluster. Within each sample, the cells are split into two groups: single cells from the cell type of interest and all other single cells. To be considered in the analysis, the gene had to be expressed in at least 10% of the single cells form one of the groups and there had to be at least a 0.25 log fold change in gene expression between the groups. This process was conducted within each sample separately, and then a meta p-value was assessed from the p-values across all samples. Multiple testing was corrected using the Benjamini-Hochberg method on the meta p-values and genes with an FDR < 0.05 were defined as cell type specific marker genes.

## Resolving cell identities of the cell clusters

Two methods were used to aid in resolving the identities of the cell type clusters. First, KEGG, Reactome, BIOCARTA, GO Biological Processes, and Hallmark gene sets were obtained from MSigDB. To identify pathways which had significant enrichment of our cell type marker genes, we used a hypergeometric test, followed by multiple testing correction with the Benjamini–Hochberg method. We also adapted the method proposed by *Zywitza et al. (2018)* to get a single cell level score for each pathway. Briefly, the expression of each gene was linearly transformed to (0,1) and the average gene expression of all genes for each gene set was computed to represent the score for that gene set. We then identified the top scoring gene sets which were representative of each cell type. The GEO accession number for the sequencing data is GSE133486.

## RNAScope fluorescence in situ hybridization (FISH)

Inguinal white adipose tissue (iWAT) from CL-treated WT mice (Jackson Laboratory, #000664) was fixed in 10% formalin overnight, embedded with paraffin, and sectioned into unstained, 5-µm-thick sections. Sections were baked at 60°C for 1 hr, deparaffinized, and baked again at 60°C for another hour prior to pre-treatment. The standard pre-treatment protocol was followed for all sectioned tissues. In situ hybridization was performed according to manufacturers' instructions using the RNAscope Multiplex Fluorescent Reagent Kit v2 (#323136, Advanced Cell Diagnostics [ACD], Newark, CA). Opal fluorophore reagent packs (Akoya Biosciences, Menlo Park, CA) for Opal 520 (FP1487001KT), Opal 570 (FP1488001KT), and Opal 620 (FP1495001KT) were used at a 1:1000 dilution in TSA buffer provided in the RNAscope Multiplex Fluorescent Reagent Kit v2. Probes targeting mm-Adrb3 (#495521, ACD) in channel 1, mm-Ucp1 (#455411-C2, ACD) in channel 2, and mm-Ppargc1b (#402131-C4, ACD) in channel four were used. Slides were mounted with ProLong Diamond Antifade Mountant with DAPI (P36966, Life Technologies). Fluorescent signals were captured with the 40x objective lens on a laser scanning confocal microscope (LSM880; Zeiss).

## Real time qPCR

Total RNA was isolated using TRIzol reagent (Invitrogen) and reverse transcribed with the iScript cDNA synthesis kit (Biorad). cDNA was quantified by real-time PCR using SYBR Green Master Mix (Diagenode) on a QuantStudio six instrument (Themo Scientific, CA). Gene expression levels were determined by using a standard curve. Each gene was normalized to the housekeeping gene 36B4 and was analyzed in duplicate. Primers used for real-time PCR are in *Supplementary file 2*.

## Western blotting

Whole cell lysate or tissue lysate was extracted using RIPA lysis buffer (Boston Bioproducts) supplemented with complete protease inhibitor cocktail (Roche). Proteins were diluted in Nupage loading dye (Invitrogen), heated at 95°C for 5 min, and run on 4–12% NuPAGE Bis-Tris Gel (Invitrogen). Proteins were transferred to hybond ECL membrane (GE Healthcare) and blotted with IL10Rα (AF-474-SP, R and D Systems), αTubulin (CP06, Calbiochem), Actin (A2066, Sigma-Aldrich).

## Cellular and mitochondrial respiration

Mitochondria were isolated from fresh tissues and immediately used in a XF24 analyzer as previously described (*Rogers et al., 2011*). Briefly, mitochondria were isolated in MSHE+BSA buffer using a 800 g/8000 g dual centrifugation method and resuspended in MAS buffer. Protein concentration was determined using a Bradford Assay reagent (Bio-Rad) and 20 µg of protein were seeded per well by centrifugation. Coupling and electron flow assays were performed as described (*Rogers et al., 2011*). For the coupling assay, basal oxygen consumption rate (OCR) was measured in the presence of 10 mM succinate and 2 µM rotenone, and after sequential addition of 4 mM ADP (Complex V substrate), 2.5 µg/ml oligomycin (Complex V inhibitor), 4 µM FCCP (mitochondrial uncoupler) and 4 µM antimycin A (Complex III inhibitor). For electron flow assays, basal OCR was measured in presence of 10 mM pyruvate (Complex I substrate), 2 mM malate and 4 µM FCCP, and after sequential addition of 2 µM rotenone (Complex I inhibitor), 10 mM succinate (Complex II substrate), 4 µM antimycin A (Complex III inhibitor) and 1 mM TMPD containing 10 mM ascorbate (Complex IV substrate).

## Statistics

All data are presented as mean ± SEM and analyzed using Microsoft Excel and Prism (Graphpad). Student's t test with Welch's correction was used for single variable comparison between two groups. One-way ANOVA followed by Dunnett post hoc test was used for multiple comparisons versus the control group. Two-way ANOVA followed by Bonferroni posttests was used to examine interactions between multiple variables. Statistical significance for CLAMS study was determined by multiple regression analysis (ANCOVA). $p < 0.05$ was considered to be statistically significant and is presented as $*p < 0.05$, $**p < 0.01$, $***p < 0.001$, or $****p < 0.0001$.

## Acknowledgements

We thank UCLA Broad Stem Cell Research Center Core for sequencing and Technology Center for Genomics and Bioinformatics for single cell sequencing. This work was supported by NIH grants R00DK114571 (PR), HL090533 (KR and PT), DK120851 and DK063491 (PT), DK104363 and UL1TR001881 (XY), and R01GM086372 (STS). A-C F is funded by the Tri-Service General Hospital, National Defense Medical Center, Taipei, Taiwan. The funders had no role in study design, data collection and interpretation, or the decision to submit the work for publication.

## Additional information

### Competing interests

Peter Tontonoz: Reviewing editor, *eLife*. The other authors declare that no competing interests exist.

### Funding

| Funder | Grant reference number | Author |
| --- | --- | --- |
| National Institutes of Health | K99 DK114571 | Prashant Rajbhandari |
| National Institutes of Health | DK063491 | Peter Tontonoz |
| National Institutes of Health | DK120851 | Peter Tontonoz |
| National Institutes of Health | HL090533 | Karen Reue<br>Peter Tontonoz |

| National Institutes of Health | DK104363 | Xia Yang |
| National Institutes of Health | UK1TR001881 | Xia Yang |
| National Institutes of Health | R01GM086372 | Stephen T Smale |

The funders had no role in study design, data collection and interpretation, or the decision to submit the work for publication.

## Author contributions
Prashant Rajbhandari, Conceptualization, Formal analysis, Funding acquisition, Methodology, Writing—original draft, Writing—review and editing; Douglas Arneson, Data curation, Formal analysis, Investigation, Methodology, Writing—review and editing; Sydney K Hart, Luis C Santos, Stephen D Lee, Investigation; In Sook Ahn, Formal analysis, Investigation, Methodology; Graciel Diamante, An-Chieh Feng, Brandon J Thomas, Laurent Vergnes, Formal analysis, Investigation; Nima Zaghari, Software, Formal analysis, Investigation; Abha K Rajbhandari, Supervision, Investigation; Karen Reue, Funding acquisition, Investigation, Writing—review and editing; Stephen T Smale, Supervision, Funding acquisition, Writing—review and editing; Xia Yang, Formal analysis, Supervision, Funding acquisition, Methodology, Writing—review and editing; Peter Tontonoz, Conceptualization, Supervision, Funding acquisition, Writing—original draft, Project administration, Writing—review and editing

## Author ORCIDs
Peter Tontonoz (iD) https://orcid.org/0000-0003-1259-0477

## Ethics
Animal experimentation: This study was performed in strict accordance with the recommendations in the Guide for the Care and Use of Laboratory Animals of the National Institutes of Health. All of the animals were handled according to approved institutional animal care and use committee (IACUC) protocol (99-131) of the University of California, Los Angeles.

## Decision letter and Author response
Decision letter https://doi.org/10.7554/eLife.49501.020
Author response https://doi.org/10.7554/eLife.49501.021

# Additional files

## Supplementary files
• Supplementary file 1. Top enriched pathways among DEGs of major cell types (FDR < 5%).
DOI: https://doi.org/10.7554/eLife.49501.014

• Supplementary file 2. QPCR primers used in this study.
DOI: https://doi.org/10.7554/eLife.49501.015

• Transparent reporting form DOI: https://doi.org/10.7554/eLife.49501.016

## Data availability
Sequencing data have been deposited to GEO (GSE133486).

The following dataset was generated:

| Author(s) | Year | Dataset title | Dataset URL | Database and Identifier |
| --- | --- | --- | --- | --- |
| Tontonoz P, Rajbhandari P, Arneson D | 2019 | Single cell sequencing of stromal vascular fraction (SVF) under B3-adrenergic agonist stimulation and mature adipocytes under cold exposure and B3-adrenergic agonist stimulation | https://www.ncbi.nlm.nih.gov/geo/query/acc.cgi?acc=GSE133486 | NCBI Gene Expression Omnibus, GSE133486 |

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
