## [Decision Letter]

**Acceptance summary:**

This study using single nuclei sequencing of adipocytes to identify a metabolically active adipocyte population that is suppressed by IL10 secreted by lymphocytes was found to be novel and important to the field. The data enhance our understanding of the role of adipose tissue immune cells in regulating adipose function and systemic metabolic homeostasis. This work will have impact in the field of immunometabolism and will provide a framework for further research on the signaling pathways that control adipocyte thermogenesis.

**Decision letter after peer review:**

Thank you for submitting your article "Single Cell Analysis Reveals Immune Cell-Adipocyte Crosstalk Regulating the Transcription of Thermogenic Adipocytes" for consideration by *eLife*. Your article has been reviewed by three peer reviewers, one of whom is a member of our Board of Reviewing Editors, and the evaluation has been overseen by Mark McCarthy as the Senior Editor. The following individual involved in review of your submission has agreed to reveal their identity: Philipp E Scherer (Reviewer #3).

The reviewers have discussed the reviews with one another and the Reviewing Editor has drafted this decision to help you prepare a revised submission.

The reviewers all agree that your work addresses an important issue, and that the manuscript advances our understanding of the role of IL10 and its receptor in adipose biology and systemic metabolism. Your nuclei scSeq method indicates substantial adipocyte heterogeneity, a most interesting observation that will be of importance to the field. However, prior to publication, a few substantive issues need clarification:

1) Has the food intake of the IL10Rα KO mice and calorie excretion been taken into account? Is it possible that much of the phenotype of these animals is simply weight loss due to less caloric input (either less intake or less retention through the gut)?

2) Analysis of an important control strain mouse carrying the adiponectin-*Cre* driver but Il10rα-/- rather than IL10rα^FL/FL^ is lacking. There are many instances of Cre drivers having toxic effects in particular cell types. Feyerabend et al. Immunity 2011 is one example. Perhaps this issue can be raised as a caveat and/or citing some other study's performance of this control with the same Cre driver line.

3) CD36 and FABP4 are also expressed in SVF, and adipocytes are sticky-how do we know these are really adipocyte populations? It is by now standard in the field to provide some independent validation of clusters defined by scRNA-seq as tSNE plots are rather artistic and vary according to chosen parameters. One approach is to use a completely different scRNA-seq platform on a replicate cohort. Immuno-histochemistry of whole adipose tissue mounts would also work, The authors should at least confirm major findings concerning population 9.

---

## [Author Response]

The reviewers all agree that your work addresses an important issue, and that the manuscript advances our understanding of the role of IL10 and its receptor in adipose biology and systemic metabolism. Your nuclei scSeq method indicates substantial adipocyte heterogeneity, a most interesting observation that will be of importance to the field. However, prior to publication, a few substantive issues need clarification:1) Has the food intake of the IL10Rα KO mice and calorie excretion been taken into account? Is it possible that much of the phenotype of these animals is simply weight loss due to less caloric input (either less intake or less retention through the gut)?

Food consumption was not different between the genotypes. This data is included in Figure 1—figure supplement 1.

2) Analysis of an important control strain mouse carrying the adiponectin-Cre driver but Il10rα-/- rather than IL10rα^FL/FL^ is lacking. There are many instances of Cre drivers having toxic effects in particular cell types. Feyerabend et al. Immunity 2011 is one example. Perhaps this issue can be raised as a caveat and/or citing some other study's performance of this control with the same Cre driver line.

As suggested, we have referenced studies using these control mice. It would not be feasible for us to repeat studies with an additional control strain within the time frame of an *eLife* revision.

3) CD36 and FABP4 are also expressed in SVF, and adipocytes are sticky-how do we know these are really adipocyte populations? It is by now standard in the field to provide some independent validation of clusters defined by scRNA-seq as tSNE plots are rather artistic and vary according to chosen parameters. One approach is to use a completely different scRNA-seq platform on a replicate cohort. Immuno-histochemistry of whole adipose tissue mounts would also work, The authors should at least confirm major findings concerning population 9.

Thank you for this suggestion. We have confirmed single cell transcriptomics from Cluster 9 using RNAScope in situ hybridization (FISH) from iWAT of saline or CL treated mice (new Figure 4F). This data directly confirms the existence and co-expression of thermogenic genes in cluster 9.

To address possible SVF contamination, we cross-matched SVF and adipocyte single cell/nuclei data (new Figure 3—figure supplement 1). A Fisher’s exact test was conducted between pairwise sets of cell type marker genes (determined by adjusted p-value < 0.05) to find cell types which had significant overlaps in their marker genes denoting transcriptional similarity. Cell types from both mature adipocyte nuclei and SVF single cells were used in this analysis and they were grouped using hierarchical clustering with tiles colored by -Log10 Bonferroni adjusted p-values. Adjusted p-values were thresholded to aid in visualization with values less than 10^-5^ set to 10^-5^. We did not find high degree of transcriptional similarity between SVF and adipocyte clusters (top). However, under stringent p-value adjustment, the transcriptomic state of adipocyte clusters 12 and 14 correlated with markers of adaptative immune cells (bottom). Thus, we cannot exclude the possibility that clusters 12 and 14 may be contaminated with immune cells.